# Wildfire, ecosystem, and climate interactions in the Early Triassic
Franziska R. Blattmann [1,2] ✉, Charline Ragon [3], Torsten W. Vennemann [1], Elke Schneebeli-Hermann [4], Christian Vérard [5], Jérôme Kasparian [3], Maura Brunetti [3], Hugo F. R. Bucher [4], Thierry Adatte [6] & Clayton R. Magill [7]

Wildfires are a key component of Earth system dynamics with respect to carbon cycling. Thus, reconstructing past wildfire dynamics is crucial for understanding potential future climate change as related to (paleo)environmental feedbacks. Here, we explore wildfire during the Early Triassic (Smithian and Spathian, ca. 250 million years ago) – a time interval characterized by scarce fire evidence, perturbation of the carbon cycle, climatic oscillations, vegetation succession and biotic radiation-extinction pulses – using polyaromatic hydrocarbons, which are an organic (geo)chemical fire indicator in sediments. Hydrocarbon abundances in shales from Spitsbergen show a prominent increase after the Smithian-Spathian boundary. Diagnostic ratios of hydrocarbons suggest that these compounds were derived from relatively unaltered biomass as opposed to soil erosion and petrogenic carbon inputs or coal combustion vis-à-vis a coincidental Siberian Trap volcanism. Our data indicates that as temperatures decline during the late Smithian, coeval hydrological conditions become less intense and changing vegetation successions become more amenable to wildfire activity. We hypothesize that changing regional wildfire regimes influenced biogeochemical cycles, potentially affecting long-term carbon sequestration. The observed coupled behavior in water-vegetation-wildfire systems amid key perturbations in Earth's history provides new insights into imminent future climate change consequences.

Wildfires are an important source of environmental disturbance. In addition to destroying habitats and emitting atmospheric aerosols, wildfires can create niche space in an ecosystem and drive rapid release of nutrients from otherwise slowly decomposing organic matter[1]. The occurrence of wildfires is therefore an important component of the Earth system regarding the nutrient and carbon cycle[1–3]. Frequencies and intensities of wildfires, and their environmental impacts operate through complex feedbacks between climate, vegetation, and landscape structure[4,5]. Consequently, paleo-wildfire activity is central to understanding (paleo-)climatic and environmental changes throughout Earth's history, particularly during major environmental crises such as mass extinctions[5]. Diagnostic molecular indicators used for reconstructing paleo- and modern wildfire, in addition to other oxidative processes (e.g., coal burning, diagenesis), are polyaromatic hydrocarbons (PAHs)[6–8]. PAHs are condensed aromatic structures created during the

incomplete combustion of fresh and degraded biomass[3]. These molecules provide insights into the dynamics of fire and the burning process, which are important for linking climate, vegetation, and carbon cycle[4].

The Permian-Triassic mass extinction (PTME) about 252 Ma is considered the most severe extinction in Earth's history[9] and was coeval with changes in wildfire dynamics[10–13]. Across the Permian-Triassic boundary, studies on charcoal and PAHs suggest fire residues were consequent to volcanism (i.e., coal burning caused by Siberian Traps intrusions), wildfire, soil erosion, or a combination of these sources, e.g.,[8,10–14]. Following the PTME and within the Early Triassic epoch spanning about 5 My (late Griesbachian to Spathian), charcoal is scarce to nonexistent in the sediment record[15,16] and no (published) PAH records exist. In this context, charcoal proxy studies have concluded that a "collapse of fire systems" led to a "charcoal gap" in sedimentary geologic archives across the Early Triassic

[1]Faculty of Geosciences and Environment, Institute of Earth Surface Dynamics, University of Lausanne, Lausanne, Switzerland. [2]Department of Geoscience, Aarhus University, Aarhus, Denmark. [3]Group of Applied Physics and Institute for Environmental Sciences, University of Geneva, Geneva, Switzerland. [4]Department of Paleontology, University of Zürich, Zürich, Switzerland. [5]Section of Earth and Environmental Sciences, University of Geneva, Geneva, Switzerland. [6]Faculty of Geosciences and Environment, Institute of Earth Sciences, University of Lausanne, Lausanne, Switzerland. [7]The Lyell Centre, Heriot-Watt University, Edinburgh, UK. ✉e-mail: Franziska.Blattmann@geo.au.dk

epoch (see refs. 17,18 and references therein). Some palynological studies have controversially reported a (re)appearance of charcoal during the Early Triassic (mostly Spathian), e.g., refs. 19,20 contradicting the widely recognized Early Triassic "charcoal gap," challenging the prevailing view of suppressed fire activity. It is not until the Middle Triassic that the first conclusive charcoal records start reappearing in the Germanic basin, e.g., ref. 15. Several hypotheses have been brought forth to explain the apparent absence of Early Triassic charcoal and, by extension, the related absence of wildfire activity. These include (a) low atmospheric oxygen concentrations preventing the ignition and spread of wildfires[21], (b) a scarcity of fuel (i.e., lack of terrestrial biomass) during this epoch[15], (c) a taphonomic bias in the charcoal and micro-charcoal record[18], and (d) a possible human bias in the published record, e.g.,[15,22,23].

In this study, we aim to circumvent the constraints of classical charcoal approaches to wildfire assessment through the extraction of PAHs from sediment samples. Three-to-six ringed PAHs were studied from the middle Smithian to the late Spathian (Olenekian) from a middle latitude outcrop of central Spitsbergen (Fig. 1). The study site contains no recognizable macro-charcoals[24]. PAH ratios and concentrations of this study site are discussed in the context of various combustion sources, such as thermal maturity, petrogenic organic carbon reworking, and pyrogenic sources. We combine modern ecological insights about fire and PAH dynamics with climate and major biomes modeling using the MIT general circulation model (MITgcm, see ref. 25 and references therein) and the BIOME4 vegetation model, respectively, with coupled biogeochemistry-biogeography parameterization (see ref. 26 and references therein). These numerical simulations provide a description at a regional scale which complements the local representation inferred from the sediment analyses. We use results of these coupled MITgcm-BIOME4 simulations covering a wide range of climate conditions[27] to quantify potential scenarios at the study site. The resulting more quantitative paleoenvironmental reconstructions[27] allow a more thorough, mechanistic examination of fire in dynamic terrestrial ecosystems.

## Results and discussion
### Thermal diagenetic overprint and petrogenic input
Past studies have demonstrated that all samples throughout the Stensiöfjellet section show an early mature range as evidenced through multiple independent parameters (e.g., conodont alteration indices (CAI = 1–1.5), palynomorphs (TAS = 4), and RockEval pyrolysis ($T_{max}$ = 430–440 °C))[24]. Molecular indices such as the methylphenanthranes index[28] feature values indicative of an early mature range (MPI = 0.34–0.65) throughout the stratigraphic section (Fig. 3A). This makes these samples optimal for molecular distribution analysis as their signal is primary and not significantly overprinted by diagenesis or catagenesis.

A steady decrease in concentration of 5–6 ringed PAHs and MPI-1 at the very base of the section in the Lusitaniadalen member (first 15 m) is likely linked to a decrease in petrogenic carbon input (Fig. 2E, F). In previous studies, Pristane/n-$C_{17}$ versus phytane/n-$C_{18}$ ratios also show a decrease in petrogenic input throughout the Lusitaniadalen Member[29]. In this context, decreasing petrogenic input is linked to regional transgression of the Arctic basin (see ref. 24 and references therein), although it is not possible to determine the exact amount of petrogenic input because concentration endmembers are not known. The ratios of 6-ringed PAHs indeno[123-cd] pyrene and benzo[ghi]perylene (IcdP/[IcdP+BghiP]) of all samples have values above 0.2 (Fig. 3B), pointing to the predominance of a pyrogenic PAH source[30]. This is further supported by the low MPI-1 (Figs. 2F, 3A)

### Pyrogenic PAH source: volcanism, soil erosion, or wildfire?
The intrusion of Siberian Trap magmatic rocks into Carboniferous-Permian coal layers is the most common theory for the cause of the PTME[31]. Volcanism and the burning of coal are associated with the occurrence of PAHs in the sedimentary record[8,12]. At Stensiöfjellet, benz(a)anthracene and chrysene ratios (BaA/(Chr+BaA)) of all samples have values above 0.35 (Fig. 3B). Following Yunker et al.[30], these benz(a)anthracene and chrysene

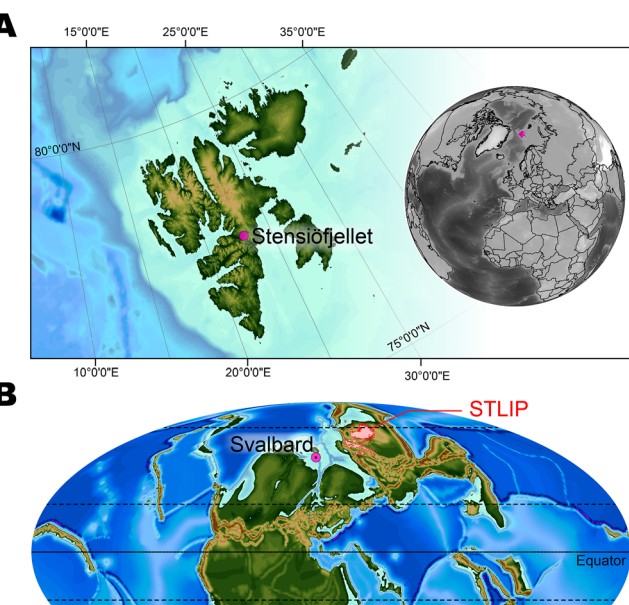

**Fig. 1 | Geographic and paleogeographic setting of Svalbard. A** Stensiöfjellet section marked in purple on a map of Svalbard with an inset map showing the location of the Svalbard Archipelago on the globe. **B** Early Triassic Pangaean continent with the location of Svalbard in purple and the Siberian Trap Large Igneous Province (STLIP) in red. Modified after Blattmann et al.[29].

ratios suggest that the PAH source at Stensiöfjellet is unlikely to be associated with burning coal deposits. Studies on mercury and mercury isotopes in the Smithian and Spathian also show that it was unlikely that extensive Siberian Trap volcanism affected the Spitsbergen region[32,33].

Past studies suggest that increased PAHs in marine sediments can be the result of an increase in soil erosion, e.g., refs. 7,12. At Stensiöfjellet, PAH levels increase at the base of the Vendomdalen Member alongside an increase in total organic carbon (TOC) (Fig. 2C–E). Both inorganic (e.g., Ti, Al concentrations)[24] and organic geochemical proxies (e.g., HI values[24,29], terrestrial-to-aquatic ratio [TAR][29]) show a decrease of terrestrial input after the Lusitaniadalen Member (Fig. 2G). This is backed by a sedimentological transition from sandy to silty shale, suggesting a shift from a proximal to a more distal marine depositional environment (Fig. 2B) that is associated with the above-mentioned transgression in the region (see ref. 24 and references therein). As the increase in PAHs shows an anti-correlation with the transgression and terrestrial input proxies (Fig. 3C), it is therefore unlikely that the PAH increase in the Spathian is derived from soil erosion.

Excluding diagenetic/catagenic overprint, petrogenic inputs, coal combustion, and soil erosion as major sources of PAHs, we hence interpret that the increase in small-to-large ringed PAHs (ug/g TOC) after the SSB is linked to an increase in wildfire activity. We argue that the "charcoal gap" in the Early Triassic is not due to a "collapse of the fire system". Instead, it is likely due to limitations of previous charcoal studies, namely (a) human bias in existing published record confirming earlier hypotheses[15,22,23] and/or (b) taphonomic bias in the charcoal record[18]. Thus, an exact reason(s) for the lack of charcoal preservation in the Early Triassic is still open to discussion (see Abu Hamad et al.[18]).

### Local, regional, or global?
The geographical extent of paleo-wildfire is central to making paleoenvironmental hypotheses about its origin(s). Following Karp et al.[6] the light molecular weight (LMW) PAHs are defined as the 3- to 4-ringed PAHs, and high molecular weight (HMW) PAHs have more than 5 rings. The LMW versus total PAHs ratio (LMW/Total)[6] shows a higher abundance (>0.5) of

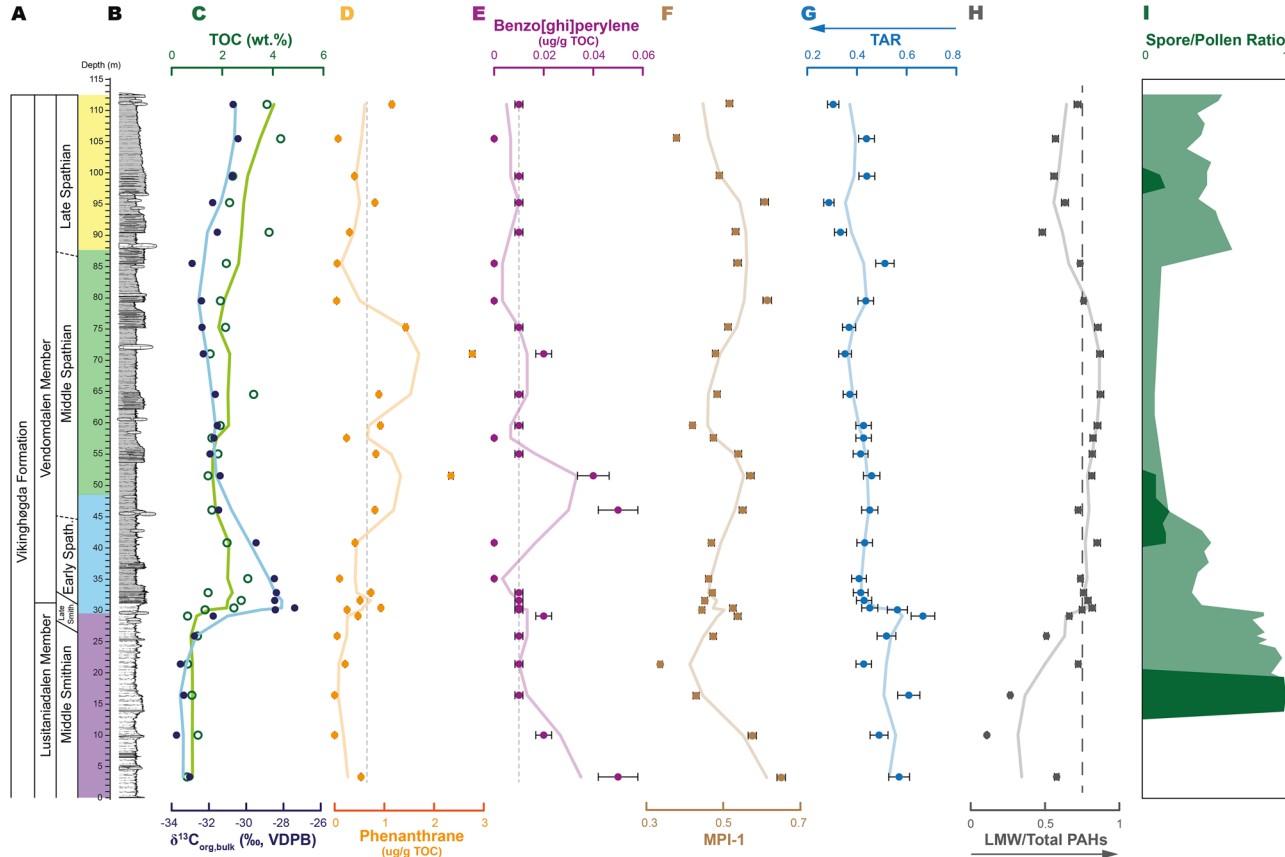

**Fig. 2 | Stratigraphy and geochemical proxies of Stensiöfjellet section. A** Stratigraphic column with ages and members of the (**B**) Stensiöfjellet section with the depth color code. The Lusitaniadalen Member consists of a light sandy shale, and the Vendomdalen Member consists of a darker silty shale[24]. **C** The $\delta^{13}C_{org}$ (VDPB, ‰) in blue, TOC (wt%) in green, (**D**) phenanthrane (ug/g TOC) in orange with the gray stippled line representing the down section average value, (**E**) benzo[ghi]perylene (ug/g TOC) in purple with the gray stippled line representing the down section average value, (**F**) methyphenanthrane index (MPI-1) in light brown, (**G**) terrestrial-

aquatic ratio (TAR) in light blue with arrow indicating decrease of terrestrial organic matter input, and (**H**) light molecular weight (LMW) over total PAHs in gray, arrow indicates increase in smoke derived PAHs with the dashed line representing the transition between residue and smoke[6]. **I** Spore/Pollen ratio reflecting the change from Lycophyte (spore producing) to Gymnosperm (pollen producing) dominated vegetation from the Svalis Dome compiled from Hochuli and Vigran[52] shown in light green, and from the Stensiöfjellet section in dark green[51]. The light shaded line in diagrams (**C–H**) represents a 3-point moving average.

LMW PAHs starting at a stratigraphic depth of 20 m (Fig. 2H) (see Supplementary Table S1 for calculation). In all samples, different trends occur among LMW and HMW PAHs (see Supplementary Fig. S1), and there are generally low concentrations of HMW PAHs and a lack of HMW PAHs such as coronene.

The high LMW/Total ratio values indicate a dominant aerosol transportation mechanism (Fig. 2H)[6]. Different trends (i.e., distinguishable slopes, see Supplementary Fig. S2) at Stensiöfjellet between LMW and HMW PAHs also point towards a difference in provinces. The lower concentrations of HMW PAHs, along with the absence of PAHs such as coronene, could indicate a substantial distance to the provenance area and/or a lack of fluvial transportation mechanism[34]. This finding also further supports the conclusion of the previous section, highlighting that Spathian PAHs increase is linked to wildfire and not soil erosion or petrogenic reworking. LMW PAHs have been documented to be transported for thousands of kilometers[35]. Therefore, we interpret the high amount of LMW PAHs present in the Spathian of Stensiöfjellet to support increasing wildfire activity in a larger region rather than a localized source only.

**Biome and climate change**

In the late Smithian, palynological studies indicate there was a widespread shift from lycophyte to gymnosperm-dominated vegetation documented from the middle latitudes to the tropics across several biomes[36–38]. On the basis of this vegetational shift, the climate is interpreted to have changed from a humid climate in the Smithian to a substantially drier climate in the

Spathian[39]. This change in the hydrologic cycle appears to be accompanied by global cooling during the late Smithian[40]. Modern and Cenozoic-based studies have demonstrated that fire, climate, and vegetation are interlinked through feedback mechanisms[4,41–44]. Yet, these crucial feedbacks are overlooked in many Mesozoic and Paleozoic geological fire reconstructions. This gap is particularly relevant given the rise of woody gymnosperms likely influenced fire behavior through altered degradation kinetics and burn intensity[45]. Additionally, extreme climate conditions and high atmospheric $p$CO₂ in the Early Triassic further obscure vegetation–fire interactions[40,46].

At Stensiöfjellet, the first pulse in PAHs occurs in the Spathian (at about 45 m depth), but smaller increases can be observed within the SSB transition already (at about 30 m depth) and throughout the Spathian (at about 70 m depth) (Fig. 2D, E). Biplotted dimethylphenanthrene (DMP) ratios, which Kappenberg et al.[47] and Karp et al.[6] developed for Cenozoic vegetation, are applied here to Mesozoic vegetation. Chemotaxonomic evidence suggests that the lipid biomarker composition of Triassic vegetation is consistent with that of contemporary taxa[48–50], allowing for the robust application of DMP ratios in this context. Our results show a compositional shift from burning predominantly herbaceous plants to more softwood input (i.e., gymnosperm) (Fig. 3D) throughout the section. The corresponding presence of retene in all samples further confirms the presence of gymnosperms[41]. The DMP trend reflecting the burnt vegetation type corresponds to the vegetational shift from lycophyte to gymnosperm-dominated seen in palynological studies from Svalbard[51–53] (Fig. 2I). This

**Fig. 3 | PAH ratio proxies for the Stensiöfjellet section.** The color scheme of the markers in all figures is based on the depth profile indicated in Fig. 2. **A** Plot of RockEval versus the methylphenanthrene index (MPI-1) values, plot modified after[28]. The error is as large as the markers. **B** Petrogrenic vs. Pyrogenic and combustion source based on ref. 30. **C** TAR vs. LMW/ Total PAHs with linear regression curve of later Smithian and earlier Spathian points (Blue and green markers) showing a moderate correlation of $r = 0.62$ and statistical significance $p = 0.03$. We report $r$ values to show the slope of the correlation. Only blue and green markers were used for regression analysis, as these capture the Smithian to Spathian transition. **D** Biplot of dimethylphenanthrenes (DMP) ratios following Karp et al.[6]. The $y$-axis error is smaller than the markers.

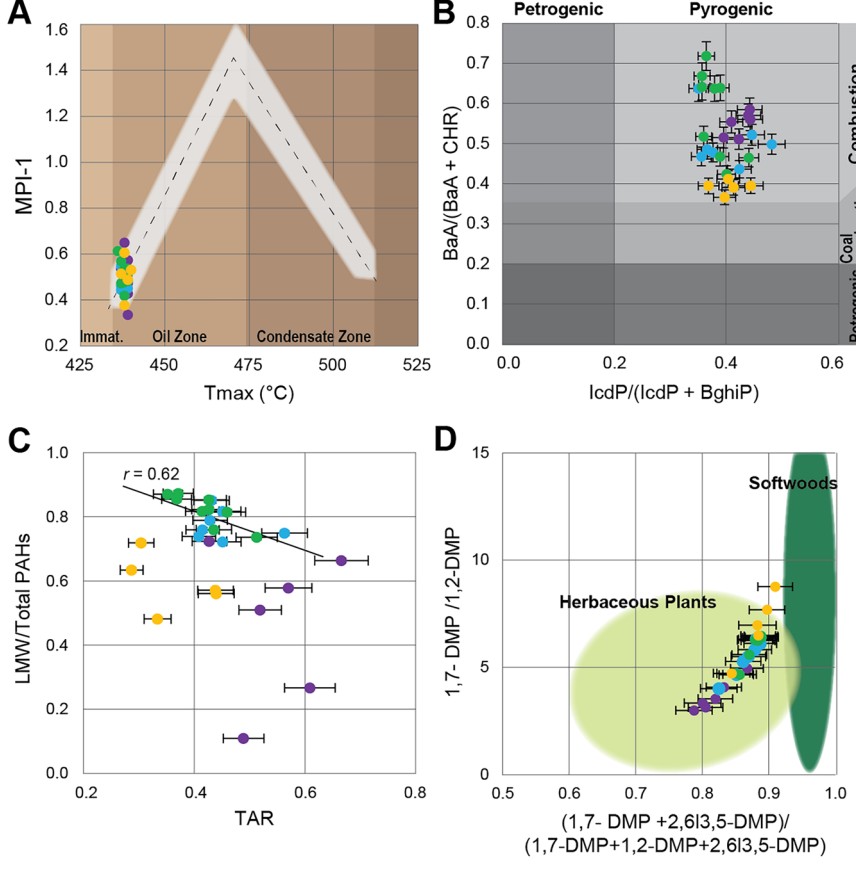

correlation between the two records further underlines the connection between changing vegetation and wildfire dynamics.

A decrease in global temperature and the weakening of the hydrological cycle are factors that have often been attributed to the vegetational change in the late Smithian[39,40]. We hypothesize that an increase in wildfire activity after the SSB is interlinked with the change in vegetation and precipitation as observed in other time periods, e.g., ref. 4. Numerical simulations using MITgcm for the Permian-Triassic paleogeography show the existence of alternative climatic states, with potential transitions between cold and hot conditions. These states differ in surface mean air temperature on the order of 10 °C and in total precipitation, displaying drier conditions in the so-called cold state[27]. Note that, despite its name, this cold state has a larger surface global temperature than the present-day climate. The offline coupling with the BIOME4 vegetation model reveals that transitioning from a hot to a cold house state would result in the replacement of forbland and dry shrubland by temperate and warm-temperate forest biomes in the direct hinterlands of the paleogeographic location of the Stensiöfjellet study site (see marked position in Fig. 4A, B).

The shift in biome from forbland and dry shrubland to temperate and warm-temperate forest agrees with the PAH and palynological proxy data mentioned above. As described by the climate simulations, this transition would also result in a local increase by ~30% in biomass content and an increase in summer precipitation relative to winter precipitation (Fig. 4C)[27]. We consider the transition between hot and cold house states comparable to the SSB transition of this study. The climate simulations, therefore, confirm that decreasing temperatures would have affected vegetation and precipitation patterns as previously observed, e.g., refs. 36–38 (Fig. 4B). In turn, these large-scale changes in vegetation and the hydrological cycle would imply a shift in wildfire activity regimes on a larger regional scale too. An increase in wildfire activity accompanied by cooler and drier climate conditions contradicts with numerous studies on more recent geological times (e.g., glacial-interglacial cycles), e.g., refs. 42,44. But, a growing number of studies

reveal that, even during glacial-interglacial times, wildfire dynamics are more complex and can vary regionally[43,54,55]. In fact, long-term wildfire activity and the associated volume of PAH cannot be estimated from the same models and proxies used for short-term fire frequency and importance prediction[56,57] that rely on fire ignition probability and the amount of available fuel, i.e., the biomass stock. On a long-term perspective, the turnover of that biomass should be taken into account through the evolution of the net biome production[58]. As a consequence, a transition to the drier and cooler climate of the cold house state, with increased seasonal precipitation intensity near the study site, may very well result in more burned biomass in the long term than the hot state at the considered latitude. Even though the specific mechanism and significance of the connection between climate and wildfire activity remains to be fully understood, this study highlights some of the unintuitive complexities in climate-wildfire-biome interactions during geological times with high atmospheric $p\text{CO}_2$ levels[46].

### Influence on biogeochemical nutrient cycling and pyrogenic carbon cycling

Studies on modern wildfire dynamics show that wildfires can affect both marine and terrestrial primary productivity by influencing the biogeochemical cycling of macro- and micronutrients, e.g., refs. 2,59. Combustion of organic matter rapidly releases soluble and thus bioavailable nutrients (e.g., nitrogen, phosphorus, and trace metals), which can be transported to the ocean via atmospheric aerosol inputs (i.e., smoke) or via increased influx of pyrogenic debris by fluvial systems[2,59,60]. Blattmann et al.[24] note an increase in marine sedimentary nutrients, particularly in phosphorus and nitrogen, but also bio-essential trace metals, at the SSB of the Stensiöfjellet section. This would further support the hypothesis brought forth in a previous study that links the paleo-wildfire record to nutrient cycling across long-term geological timescales[21]. However, further studies spanning a broad geographic range and multiple time intervals are needed to more robustly evaluate the correlation between wildfire dynamics and nutrient cycling.

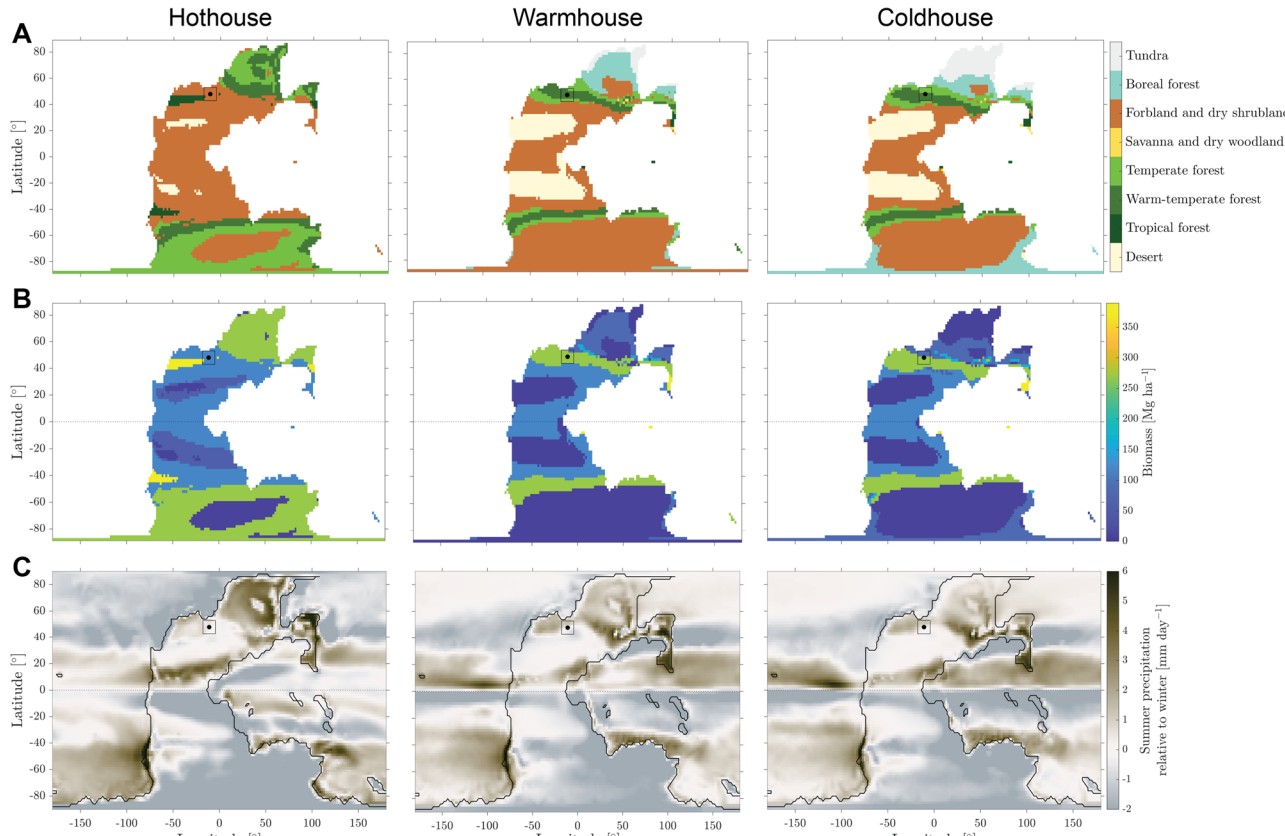

**Fig. 4 | Model simulations with MITgcm and BIOME4 of a hothouse, warmhouse, and coldhouse following Ragon et al.[27]. A** Biome map with legend to the right, (**B**) biomass concentration with the legend to the right, and (**C**) map of the summer precipitation relative to winter precipitation in mm/day of the hot, warm, and cold house states with the legend to the right. The study site is denoted by a dot within a 1500-km-sided square.

The combustion of terrestrial organic matter during wildfires leads to emissions of carbon dioxide, trace gases, and aerosols into the atmosphere via the oxidation of organic carbon. However, the incomplete combustion of biomass also generates pyrogenic carbon (PyC). Many PyC compounds are relatively recalcitrant in comparison to labile organic matter, as PyC decomposes more slowly than the bulk organic matter. As PyC accumulates in soils and sediments, the rapid carbon emissions of the wildfire are subsequently partly offset[3]. Recent studies have shown that the PyC makes up a large portion of terrestrial organic carbon export (15% or more of total terrestrial organic carbon flux)[61] with a strong dependency on the biome[61]. Depending on the wildfire extent, the production of PyC might hence act as a carbon sequestration mechanism, e.g., ref. [62]. PyC cycle feedback mechanisms observed in contemporary records suggest that fluctuating paleo-wildfire frequency likely represents an underexplored component of the carbon cycle across geological time. This is particularly relevant during pivotal intervals of major carbon cycle perturbation and rapid environmental change, such as the Smithian–Spathian. Further studies are needed to better understand the long-term connection between wildfire dynamics and carbon cycling across geological timescales.

## Materials and methods
### Study site and sampling
The Stensiöfjellet site is located in central Spitsbergen (Lat = 78.283058°N; Long = 17.716164°E) (Fig. 1). It encompasses the sandy shales of the Lusitaniadalen Member and the dark silty shales of the Vendomdalen Member, which are part of the Vikinghøgda Formation (Fig. 2A, B)[24]. A hiatus occurs at the members boundary seen across all of Svalbard[63]. A detailed description of the section and regional geology can be found in Blattmann et al.[24], Vigran et al.[63], and Mørk et al.[64].

The Stensiöfjellet section spans the upper middle Smithian (*Owenites koenei* beds), the late Smithian *Wasatchites* beds, and the entire Spathian[51,65]. The SSB definition used in this study is based on ammonoid zones[66], noting that no official definition exists and that alternative schemes have been applied in other studies (e.g., Zhang et al.[67]).

Twenty-seven samples were collected at the Stensiöfjellet section specifically for biomarker analysis. The ~1 kg samples were collected at approximately 5 m intervals and at the SSB at 1–2 m intervals. To avoid contamination, all samples were handled with care and wrapped in ashed aluminum foil.

### RockEval pyrolysis
A Rock-Eval six was used to determine $T_{max}$ on powdered whole rocks (see Supplementary Table S2). The samples were pulverized using an agate mortar. The IFP 160000 standard was used to calibrate all measurements with an instrumental precision of 2 %[68].

### Biomarker extraction and isolation
Freeze-dried sediments were crushed into a homogenous powder with a solvent-cleaned agate mortar and pestle. Then, sediment powders were extracted via accelerated solvent extraction (Dionex ASE 350 system) using dichloromethane (DCM) and methanol (MeOH) (9:1 vol/vol) in three cycles at 100 °C (10.3 MPa) with a static extraction time of 5 min. Resultant total lipid extracts (TLEs) were air-dried and then chromatographically partitioned into three fractions with "flash" open column separation. Individual flash columns—crafted from ashed 150 mm soda-lime glass Pasteur pipettes with a small plug of glass wool in the neck—were made with ~0.5 g of deactivated silica gel[69]. To each column, a sediment TLE was introduced in ~0.2 mL hexane and then eluted through the column with 4 mL fractions of hexane (F1), DCM (F2), and MeOH (F3).

## Molecular analysis

Lipid biomarkers, viz. PAHs in F1 were characterized by gas chromatography-mass spectrometry (GC–MS; Thermo Scientific TRACE 1310 [GC] with coupled ISQ LT [MS]) by splitless injection of 1 μL aliquots of individual lipid fractions onto a 60-m DB-5 fused-silica column (0.25 mm × 0.25 μm). The GC oven was programmed to inject at 60 °C and hold for 2 min, ramp at 10 °C/min to 150 °C, ramp at 4 °C/min to 300 °C, followed by an isothermal hold of 20 min. Procedural blanks and bracketing standards were run to monitor contamination and background interferences.

PAH compound identification was made via comparison with the QTM PAH Mix (CRM47930) standard through selective ion monitoring (SIM) channels $m/z$ 178 (phenanthrene, anthracene), 192 (methylphenanthranes), 202 (fluoranthene, pyrene), 206 (dimethylphenanthranes), 228 (benz[a]anthracene, chrysene), 252 (benzo[b]fluoranthene, benzo[k] fluoranthene, benzo[a]pyrene, benzo[e]pyrene, perylene), and 276 (indeno[1,2,3-cd]pyrene, benzo[ghi]perylene)[6,70] in conjunction with the NIST20 electron ionization spectral library (see Supplementary Figs. S3–5 and Supplementary Table S3). Individual PAHs were quantified using SIM channels with response factors determined from respective compound 5-point calibration curves. An overview of all analyzed PAH is available in the appendix. All errors were calculated using the coefficient of variation. The coefficient of variation was calculated from replicate measurements of the 16-component QTM PAH Mix certified reference standard (Supelco CRM47930). Propagated uncertainty was calculated through relative standard deviation (RSD) for combined instrumental and analytical uncertainties of QTM PAH Mix replicate measurements ($n = 14$; RSD precision <3%).

## Data analysis

We used change-point analysis to identify thresholds in our datasets, detecting pulses as significant shifts relative to background trends. This approach is ideal for continuous numeric data with relative (as opposed to absolute) age constraints (c.f., Qian et al.[71]). Our PAH records show distinct transitions at the Smithian–Spathian Boundary (~30 m) and the Early–Middle Spathian (~46 m) at the 95% confidence level ($p < 0.05$), consistent with shifts in system state[72]. Limited Early Triassic PAH records, however, preclude evaluation against longer-term trends.

## Numerical simulations

The three steady states obtained using the MIT general circulation model coupled with the vegetation model BIOME4 are described in detail in ref. 27. These states are characterized by different climatic variables, in particular surface air temperature, precipitation, and vegetation cover on land, as shown in Fig. 4. The BIOME4 model estimates the global steady state of the vegetation distribution and resulting major biomes corresponding to the simulated climate at each location on land[26].

## Data availability

All geochemical data from this study are accessible on Zenodo (https://doi.org/10.48657/djz0-sc60). The data of the numerical simulations used in this study were generated using the MIT General Circulation Model (MITgcm; http://mitgcm.org/, https://github.com/MITgcm/MITgcm, versions c67f and c67j) and BIOME4, both of which are openly available. The coupled MITgcm-BIOME4 configuration, relevant data, and numerical results are in Ragon et al.[27].

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

## Acknowledgements
This work is supported by the Swiss National Science Foundation (Project n° CRSII5-180253 to H.F.R.B., T.W.V., M.B., Urs Schaltegger, and CRSII5-180253/2 Mobility Grant to F.R.B. and T.W.V.). Tessa Plint is thanked for all her support in the laboratory. Fieldwork permits were applied for under the number RiS-ID 11491, and the authors thank the Governor of Svalbard for granting us these permits. Three anonymous reviewers are thanked for their helpful comments.

## Author contributions
F.R.B. conceived the study, acquired funding, performed formal analyses, modeling, and data curation, and led the writing and editing of the manuscript. C.R. contributed to modeling, data investigation, validation, and manuscript editing. T.W.V. provided validation, supervision, funding acquisition, and manuscript editing. E.S.-H. contributed validation and manuscript editing. C.V. contributed to paleogeographic reconstruction, validation, and manuscript editing. J.K. and M.B. contributed to conceptualization, modeling, manuscript critique, and M.B. to funding acquisition. H.B. participated in the fieldwork, contributed to funding acquisition, and manuscript editing. T.A. contributed methodology and manuscript editing. C.R.M. contributed conceptualization, methodology, resources, data curation, validation, supervision, and manuscript editing. All authors contributed to the analysis and interpretation of results, discussed the findings, and provided input on the final manuscript.

## Competing interests
The authors declare no competing interests.
