## [Transparent Peer Review file · Communications Earth & Environment]

Burning questions - wildfire, ecosystem, and climate interactions in the Early Triassic

Corresponding Author: Dr Franziska Blattmann

Version 0:

Decision Letter:

Dear Dr Blattmann,

Your manuscript titled "Burning questions: wildfire, ecosystem, and climate interactions in the Early Triassic" has now been seen by 3 reviewers, whose comments are appended below. You will see that they find your work of some potential interest. However, they have raised quite substantial concerns that must be addressed. In light of these comments, we cannot accept the manuscript for publication, but would be interested in considering a revised version that fully addresses these serious concerns.

We hope you will find the reviewers' comments useful as you decide how to proceed. Should additional work allow you to address these criticisms, we would be happy to look at a substantially revised manuscript. If you choose to take up this option, please either highlight all changes in the manuscript text file, or provide a list of the changes to the manuscript with your responses to the reviewers.

When resubmitting, please provide a point-by-point response to the reviewers' comments. Please submit your responses as a separate file, distinct from your cover letter where you can add responses to the Editors' comments that you do not want to be made available to the reviewers. Word files are preferred. We recommend that any figures, tables or graphs that are included in the response to reviewers are also included in the main article or Supplementary Information.

If the revision process takes significantly longer than three months, we will be happy to reconsider your paper at a later date, as long as nothing similar has been accepted for publication at Communications Earth & Environment or published elsewhere in the meantime.

Please use the following link to submit your revised manuscript, point-by-point response to the reviewers' comments with a list of your changes to the manuscript text (which should be in a separate document to any cover letter), a tracked-changes version of the manuscript (as a PDF file) and any completed checklist:

Link Redacted

Please do not hesitate to contact us if you have any questions or would like to discuss the required revisions further. Thank you for the opportunity to review your work.

Best regards,

Carolina Ortiz Guerrero, Ph.D.
Associate Editor
Communications Earth & Environment

EDITORIAL POLICIES AND FORMAT

If you decide to resubmit your paper, please ensure that your manuscript complies with our editorial policies and complete and upload the checklist below as a Related Manuscript file type with the revised article:

Editorial Policy [Policy requirements](https://www.nature.com/documents/nr-editorial-policy-checklist.pdf) (Download the link to your computer as a PDF.)

- Behavioural and social science
- Ecological, evolutionary & environmental sciences
- Life sciences

<https://www.nature.com/documents/nr-reporting-summary.zip>

For your information, you can find some guidance regarding format requirements summarized on the following checklist: (<https://www.nature.com/documents/commsj-phys-style-formatting-checklist-article.pdf>) and formatting guide (<https://www.nature.com/documents/commsj-phys-style-formatting-guide-accept.pdf>).

REVIEWER COMMENTS:

Reviewer #1 (Remarks to the Author):

I accept the paper with minor changes. Mainly a few refs as missing but otherwise a very well written and concise paper. The data is of good quality and the interpretations are accurate with current knowledge on PAHs in P/Tr and recovery and refs to modern fire regims and other extinction and recovery events like Triassic/Jurassic.

Reviewer #2 (Remarks to the Author):

The paper presents some interesting data on PAHs recovered from the Early Triassic (Smithian and Spathian substages). However, I think the paper is poorly structured and this significantly impacts on the readability of the paper. As an example of this the key results are first cited in the introduction and then underpinning data about low thermal maturity etc of the site is presented as the first figure within the results section (figure 3). The paper then transitions back to talking about the key data presented in figure two in the next section of the results and discussion.

To provide a clearer and a more logical framework for the study the introduction needs to be expanded so the different sections of the ms can be fully introduced. The first paragraph of the introduction needs to be underpinned by more citations. Why are the references cited in line 87 controversial and who thinks these records are controversial? The sentence presented in line 96-97 as written suggest that the PAH data will be compared to charcoal analysis.

More specifically in terms of the data and data presentation I have the following comments:

The paper makes claims linked to higher abundance of LMW PAHs (line 171) and pulses of PAHs occurring in the Spathian (line 192). How have these shift in abundance and pluses been determined, are the events described actually a statistically significant deviation from their respective long-term mean or are they just part of the natural variation?

In figure two it isn't clear what the grey line plotting the variation in PAHs actually is or how it was derived. These details need to be included. Information on what the error bars on the individual data points are also need to be included.

The figure legend for figure three doesn't match the with the plots, there is a linear regression within plot 3C but the legend describes this in plot 3D. It is unclear why only the green and blue data points chosen for this analysis or why this analysis was done. Also the statistics need to be reported in full rather than just an isolated r value. I think it might be better to report the R2 rather than r vaule or explain why the r is reported. Is the relationship significant what is the P value?

For figure 3D what information is used to constrain what is classified as herbaceous and what is softwood. If this designation comes from the modern world are do these hold true for the early Mesozoic angiosperm free world? The data presented in figure 3D seems to be at odds with what is described in the text (line 194-195 and lines 197-198) as all but one of the data points clusters in the field designated as being indicative of herbaceous plants.

Palynology studies inferring vegetation change are mentioned (line 198) but not cited and no data is presented. As such it is difficult to comment on if there is a correlation or not between the datasets and what this means or does not mean for wildfire dynamics. Are the authors inferring a statistical or stratigraphic correlation?

The model work doesn't feel integrated into the paper I think this is because the wider aspects of the study and wildfire dynamics are not fully discussed in the introduction. Line 214-215 suggest that the figure presented as Figure 3B is evidence supportive of decreasing temperature impacting on precipitation and vegetation. This seems at odds with the the data presented in the figure and the description within the legend.

Given that the authors present data from one section and it isn't clear how these data map onto the broader area at this time interval or if the fire activity they have detected is substantive or not the discussion of broader influences of wildfires (Line 225-237) should be softened.

The final paragraph of the paper talks about a topic (PyC) not previously mentioned and this seems like a rather strange way to close out a paper.

Reviewer #3 (Remarks to the Author):

Overview:

The authors present PAHs and alkanes records spanning Smithian-Spathian substages, ca. 250 million years ago. The source of PAHs and the geographical extent of paleo-wildfire were discussed in detail. The manuscript present a novel interpretation of wildfire intensities shifted together with changing vegetation and precipitation patterns, as well as regional organic carbon cycle. However, the discussion about relations between wildfires and climatic conditions need to be more focused to improve readability and logicity for readers. The method should be more detailed to enhance the reader's understanding of the dataset and results. Both figures in text and supplemental figures require minor revision. Please see specific comments below.

Specific comments:

Line 75: Please add some description about the origin of PAHs and their climatic significance during the mass extinction events if word count allows.

Line 82: It is better to refer the stages or conodont zones here, so that the reader can compare between different sections.

Line 113: Please give specific values for the CAI, Ro% and MPI, and then discuss the thresholds for maturity (i.e. which value for immature/mature) to improve the reading interest of non-professional readers.

Line 115: In many publications, the plot Pr/nC17 vs. Ph/nC18 suggested the origin of organic matter or OM type, and the authors could cite published papers instead of manuscript under review. Pr/nC17 and Ph/nC18 ratios are also thought to decrease with increasing thermal maturity. However, this relationship only holds if the decomposition of the side chain of chlorophyll-a is the only or at least the predominant source of these isoprenoids. Please discuss in detail that why the plot Pr/nC17 vs. Ph/nC18 can be the indicator of maturity and what type of OM in this study.

Line 121: There are points in Figure 2E representing values of 0 (or nearly?) that should not be include in the graph. Please provide more reasonable interpretation of PAHs profile.

Line 123: Please include an interpretation of why the Pristane/n-C17 and phytane/n-C18 ratios can indicate petrogenic input. Additionally, please add this proxy to Figure 2 to illustrate the changes in petrogenic input across the section.?

Line 129: Petrogenic input can be assessed by changes in alkyl PAHs, with various indicators available to quantify alkyl PAHs for evaluating petrogenic input.

Line 134: delete e.g. and cite publications directly.

Line 139: Please consider another possible source that PAHs produced by regional wildfire and transport via aerosols.

Line 142: "PAH levels" specifically refers to which indicators in Figure 2, and it seems that the figure does not match the description.

Line 152: The source of wildfire is discussed by excluding potential sources; please provide more concrete and direct evidence..

Line 162: The first sentence of this paragraph mentions biogeochemical cycles, please add explanations about impacts on biogeochemical cycles to increase the manuscript's readability by creating connections between the first sentence and subsequent paragraphs.

Line 164: The interpretation of LMW and HMW could add in supplementary table 2.

Line 171: LMW is very common in Phanerozoic strata, so it is also considered as petrogenic input.

Line 172: What does "distinct trends between LMW and HMW" mean ?

Lines 173-180: The explanation of high LMW PAHs and low HMW PAHs is somewhat vague. Generally, the formation temperature or geothermal temperature may not be sufficient to produce HMW PAHs. However, the loss of HMW PAHs during long-distance transport could be one possible explanation. It is better to interpret PAHs transportation in conjunction with conclusive evidence or publications about increased wildfire activity in a large region, along with the wind systems that transport these PAHs to the study area.

Line 180: Please add wildfire evidence or sites in the larger region and how they transport to the study area.

Line 184: Vegetation types exhibit significant regional characteristics. Please emphasize in this paragraph that changes in regional vegetation types can indicate drier climate. Excepting changes in vegetation type, are there other evidences (i.e. thorns or mud crack) that suggest drought conditions?

Line 189: Many studies have examined the relationships among fire, climate, and vegetation; however, it would be valuable if the authors could offer a new perspective on these three aspects.

Line 193: In Figure 2, there was also an increase at about 70 m depth, and why this increase not be included in the

discussion?

Line 195: Please point out the timing of vegetation transition and the appearance of the retene in the profile.

Line 197: Are the molecular evidence and vegetation shifts consistent over time? At what specific time or depth in the profile do they align? Clarifying this will help the reader understand the sequence of events and the relationships among fire, climate, and vegetation.

Line 210: Is there fossil evidence about the increase in biome diversity locally? If so, please cite them.

Line 215: Figure 3B is refer to the source analysis of PAHs, please re-cite figure.

Lines 225-236: This part is speculative and lacks precise, direct evidence regarding nutrient and carbon cycles. I suggest shortening this part and merging it with the model analysis above to enhance the completeness of the article.

Line 245: Wildfires are thought to be a major source of carbon dioxide released by forests into the atmosphere. Large and intensive wildfires can lead to forest ecosystems shift from carbon sinks to carbon sources. However, carbon sequestration mechanism in study area seems to lack of direct evidence. Additionally, the article does not address the frequency of fires, making it difficult to definitively conclude that carbon sequestration is occurring. It may be more appropriate to conclude that wildfires can have a significant impact on the global carbon cycle, although further data and models are needed to fully explain this impact.

Line 252: The method should be more detailed to enhance the reader's understanding of the dataset and the identity of results.

Line 254: Please add the paleogeographic latitude and longitude, regional climate conditions and section description in the method. If publications have detailed descriptions of the section, please cite them in the method.

Line 260: Please describe the maturity proxies and the results of RockEval in detail. It is recommended to put the maturity proxies and results in the method part or supplementary file.

Line 280: Please add description of the compound identification and references for the PAHs identification in supplementary file.

Line 284: Please add one paragraph about detailed steps of the model and a description of the model results.

Figures :

Figure 2: Please add the significance of the arrows on the horizontal axis. add the lithology legend.

Figure 3: Please add a significance P-value test.

Figure 4: Please label A, B, C that match with text.

Supplementary Figure 1: Please enlarge the horizontal and vertical coordinates.

Supplementary Figures 2-4: Please add identification of PAHs. The peak of $m/z = 276$ could not be identified in the figure so it is necessary to describe the identification in the text.

Supplementary Figure 5: The 0 value could not be shown in the profile, especially HMW PAHs profiles.

Communications Earth & Environment is committed to improving transparency in authorship. As part of our efforts in this direction, we are now requesting that all authors identified as 'corresponding author' create and link their Open Researcher and Contributor Identifier (ORCID) with their account on the Manuscript Tracking System prior to acceptance. ORCID helps the scientific community achieve unambiguous attribution of all scholarly contributions. You can create and link your ORCID from the home page of the Manuscript Tracking System by clicking on 'Modify my Springer Nature account' and following the instructions in the link below. Please also inform all co-authors that they can add their ORCIDs to their accounts and that they must do so prior to acceptance.

If you experience problems in linking your ORCID, please contact the Platform Support Helpdesk.

Version 1:

Decision Letter:

Dear Dr Blattmann,

Your revised manuscript titled "Burning questions: wildfire, ecosystem, and climate interactions in the Early Triassic" has now been seen by our reviewers, whose comments appear below. In light of their advice we are delighted to say that we are happy, in principle, to publish a suitably revised version in Communications Earth & Environment.

We therefore invite you to revise your paper one last time to address the remaining concerns of our reviewers. At the same time we ask that you edit your manuscript to comply with our format requirements and to maximise the accessibility and therefore the impact of your work.

EDITORIAL REQUESTS:

****Please take care to match our formatting and policy requirements. We will check revised manuscript and return manuscripts that do not comply. Such requests will lead to delays. ****

SUBMISSION INFORMATION:

OPEN ACCESS:

Communications Earth & Environment is a fully open access journal. Articles are made freely accessible on publication. For further information about article processing charges, open access funding, and advice and support from Nature Portfolio, please visit <https://www.nature.com/commsenv/open-access>

Link Redacted

Best regards,

Alireza Bahadori, PhD
Associate Editor
Communications Earth & Environment
Consulting Editor
Communications Sustainability

REVIEWERS' COMMENTS:

Reviewer #2 (Remarks to the Author):

Thank you for responding to my comments and providing the detailed response letter outlining how you have responded to questioned raised in my initial review of your paper. I think some of the text within your response letter needs to be within the body of the paper. For example the discussion on how you handled the data to detect pulses of PAHs is missing from the text. This needs to be included. If you are basing the analysis on shifts from the section average I think it might be worth thinking about replotting your data as difference from the average this would highlight the nature of the PAH spike. Another minor example of information that would enhance the readability of the document is the reason why the the studies on line 89 are controversial.

The introduction still feels a little bit underdeveloped and that is still impacting on readability. The modelling still feels more like a bolt onto the ms rather than an integral part of the study. I would strongly encourage the authors to take advantage of the greater word limit for Communications Earth & Environment to provide a more extensive set up the paper and provide a more detailed discussion of some of the relationships you have found. For example I think it would be beneficial to explore in

more detail the mixing model presented on lines 205-208. The authors highlight in the response this is backed up figure four but again this isn't articulated in the paper.

I think many of the issues raised by reviewer three and responded to in the cover letter would also be better folded into the revised ms rather than being in the response.

I recognise that the structuring of a paper is rather subjective but I'm not sure having the main results of the paper being presented in the introduction (figure 2) is the logical way to structure the paper. The main purpose of the citing of the figure in the introduction is to cite the sedimentary log. If the authors wish to do this wouldn't it be better to combine the log with the location map to have one complete introduction figure and then have the actual data embedded in the results section?

Reviewer #3 (Remarks to the Author):

The manuscript has been revised according to the comments and has been reorganized to improve its logic and completeness. Based on the author's response, there are still four comments that require further clarification in the text:

1. The response to comment 2 (Line 85) still needs to specify the biostratigraphy. If there are regional differences, it should be indicated which regional biostratigraphy belongs to and the corresponding international stratigraphic age. Additionally, if word count allows in the methods section, some information on stratigraphy and sedimentology could be added, as ref. 24 and 62 do not present absolute chronostratigraphy or stratigraphy research.

2. The response to comment 5 (Line 121): Can I understand that these very low values are actually background values? If so, the author could consider using the detection limit as the background value. This would make the discussion of PAH concentrations as evidence of fire more reasonable.

3. The response to comment 9 (Lines 117–137 and 139-167): The biological source is also possible source, especially LWM PAHs. Considering the need for rigor, this possibility needs to be ruled out, e.g. Krauss, M., Wilcke, W., Martius, C., Bandeira, A. G., Garcia, M. V., & Amelung, W. (2005). Atmospheric versus biological sources of polycyclic aromatic hydrocarbons (PAHs) in a tropical rain forest environment. *Environmental Pollution*, 135(1), 143-154.

Daisy, B. H., Strobel, G. A., Castillo, U., Ezra, D., Sears, J., Weaver, D. K., & Runyon, J. B. (2002). Naphthalene, an insect repellent, is produced by *Muscodor vitigenus*, a novel endophytic fungus. *Microbiology*, 148(11), 3737-3741. Although studies on its biological origin are modern researches, it cannot be completely ruled out that ancient organisms are not a source of PAHs in the strata.

4. The response to comment 11: I agree that the excluding potential sources in this ms is logical deduction. However, if the process of elimination is needed to determine the source of PAHs, all possible sources of PAHs should be excluded, as mentioned above.

** Visit Nature Portfolio's author and referees' website at www.nature.com/authors for information about policies, services and author benefits**

Dear Dr. Ortiz Guerrero,

Thank you for your consideration of our manuscript titled "Burning questions: wildfire, ecosystem, and climate interactions in the Early Triassic". We appreciate the time and effort you and the reviewers have taken to provide detailed and constructive feedback.

We acknowledge the reviewers' concerns and agree that several areas of the manuscript required clarification and structural improvement. In response, we have undertaken revisions to address all comments carefully and systematically. These include a restructuring of the manuscript to improve readability and flow, expanded discussion of proxy interpretations and uncertainties, clarification of methodological approaches, figure revisions and the inclusion of additional references where appropriate.

We are submitting a revised version of the manuscript along with:

- A detailed, point-by-point response to each reviewer comment.
- A tracked-changes version of the manuscript.
- A clean version of the revised manuscript.
- Figure, supplemental figures and tables

We hope that the revisions meet the expectations of the reviewers and the editorial team, and that the improved manuscript will now be suitable for publication in Communications Earth & Environment. We are aware that the number of citations is exceeding the limit given by Communications Earth & Environment. We would like to point out these have been added due to the reviewer request.

Please do not hesitate to contact us should you require any additional information.

On behalf of all co-authors,

Franziska R. Blattmann, PhD

All line comments made below refer to the tracked changes manuscript.

REVIEWER COMMENTS:

Reviewer #1 (Remarks to the Author):

I accept the paper with minor changes. Mainly a few refs as missing but otherwise a very well written and concise paper. The data is of good quality and the interpretations are accurate with current knowledge on PAHs in P/Tr and recovery and refs to modern fire regims and other extinction and recovery events like Triassic/Jurassic.

We thank Reviewer 1 for their positive evaluation of our manuscript and for recognizing the quality of the data, clarity of writing and relevance of our interpretations within the current understanding of PAHs across the Permian-Triassic boundary. We are particularly grateful for the comments acknowledging the paper's conciseness and the connection to both ancient and modern fire regimes, as well as parallels with other extinction and recovery events such as the Triassic/Jurassic transition.

We are pleased to have addressed these final refinements (see below).

1. Line 52: plural PAHs

Thank you for pointing out this mistake. It has been corrected.

2. Line 75: references to include here are

Relevant references have been added to this sentence. Please see Line 74-76.

3. Line 85: coal gap

We would like to clarify that charcoal and coal are not synonymous. In this paper, our discussion refers to the "charcoal gap" rather than to coal deposits.

Reviewer #2 (Remarks to the Author):

The paper presents some interesting data on PAHs recovered from the Early Triassic (Smithian and Spathian substages). However, I think the paper is poorly structured and this significantly impacts on the readability of the paper. As an example of this the key results are first cited in the introduction and then underpinning data about low thermal maturity etc of the site is presented as the first figure within the results section (figure 3). The paper then transitions back to talking about the key data presented in figure two in the next section of the results and discussion. To provide a clearer and a more logical framework for the study the introduction needs to be expanded so the different sections of the ms can be fully introduced.

We thank Reviewer 2 for their valuable and constructive feedback. We sincerely apologize for the structural issues in the original submission that impacted the clarity and flow of the manuscript. We appreciate the Reviewer's specific observations regarding the placement of key results and supporting data.

In response, we have carefully restructured the manuscript to improve readability and logical flow. Specifically, we have adjusted the sequence of the Results and Discussion sections to

ensure that foundational information (such as thermal maturity data) is presented before interpreting the PAH concentrations and patterns (removing lines 113-115). We have also revised the introduction to avoid prematurely discussing key results and introducing the various discussed aspects (Line: 76-78; 105-107; 112-114).

1. The first paragraph of the introduction needs to be underpinned by more citations.

See comment 2 of Reviewer 1. See References 1-8

2. Why are the references cited in line 87 controversial and who thinks these records are controversial?

The references cited in line 87 have been considered controversial primarily within the palynological community because they report an abundance of charcoal in intervals where previous studies have consistently noted an absence (commonly referred to as the "charcoal gap"). These findings challenge the prevailing interpretation of widespread suppression of fire activity during the Early Triassic recovery period. As such, they have sparked debate regarding their representativeness.

3. The sentence presented in line 96-97 as written suggest that the PAH data will be compared to charcoal analysis.

We apologize for the misleading phrasing (Lines 99 -100), which unintentionally suggested that the PAH data would be directly compared to charcoal analysis. We have revised the sentence to clarify our intended meaning and to ensure that the reader can follow the logical flow of the manuscript without confusion.

More specifically in terms of the data and data presentation I have the following comments:

4. The paper makes claims linked to higher abundance of LMW PAHs (line 171) and pulses of PAHs occurring in the Spathian (line 192). How have these shift in abundance and pluses been determined, are the events described actually a statistically significant deviation from their respective long-term mean or are they just part of the natural variation?

We use change-point analysis to detect thresholds in our datasets and thus identify 'pulses' or events (i.e., a shift in average variance) distinct from natural/background trends and fluctuation. This approach is ideal for continuous numeric data such as ours with relative (as opposed to absolute) age constrains (c.f., Qian et al. 2003). Within our own datasets, associated PAH datasets demonstrate consistent shifts in median value at the Smithian-Spathian Boundary (~30m depth) and Early-to-Middle Spathian transition (~46m depth) at the 95% confidence level (p-value < 0.05) that are consistent with distinct transitions in system state (Aminikhangahi & Cook 2016). Unfortunately, lacking additional records of PAHs from the Early Triassic epoch preclude our ability to evaluate variance against longer-term trends before or after our record's coverage. Certainly, something to work toward going forward, though!

Qian et al. 2003. Two statistical methods for the detection of environmental thresholds. *Ecological Modelling* 166:87-97

Aminikhanghahi S, Cook DJ. A Survey of Methods for Time Series Change Point Detection. *Knowl Inf Syst.* 2017 May;51(2):339-367. doi: 10.1007/s10115-016-0987-z. Epub 2016 Sep 8. PMID: 28603327; PMCID: PMC5464762.

5. In figure two it isn't clear what the grey line plotting the variation in PAHs actually is or how it was derived. These details need to be included. Information on what the error bars on the individual data points are also need to be included.

We thank Reviewer 2 for pointing out the lack of clarity regarding the grey line and error bars in the Figures. This was an oversight on our part, and we appreciate the opportunity to correct it. We have now added an explanation of how the grey line (representing the three-point moving average in PAH concentrations) was derived. Additionally, we have clarified the error bars associated with the individual data points and included this information to ensure transparency. (Lines: 320–324; Line:580)

6. The figure legend for figure three doesn't match the with the plots, there is a linear regression within plot 3C but the legend describes this in plot 3D. It is unclear why only the green and blue data points chosen for this analysis or why this analysis was done. Also the statistics need to be reported in full rather than just an isolated r value. I think it might be better to report the R² rather than r vaule or explain why the r is reported. Is the relationship significant what is the P value?

Thank you for pointing out the mistake in the figure legend. This has now been corrected. We have also added an explanation to the figure legend and included additional statistical information to support the interpretation and improve clarity for the reader. (Lines: 582-592)

7. For figure 3D what information is used to constrain what is classified as herbaceous and what is softwood. If this designation comes from the modern world are do these hold true for the early Mesozoic angiosperm free world? The data presented in figure 3D seems to be at odds with what is described in the text (line 194-195 and lines 197-198) as all but one of the data points clusters in the field designated as being indicative of herbaceous plants.

We base the classification of herbaceous versus softwood input in Figure 3D on a combination of phylogenetic and chemotaxonomic evidence. While the precise taxonomic composition of Smithian–Spathian vegetation remains uncertain, molecular phylogenies calibrated with fossil ages (Rai et al. 2008), along with lipid biomarker chemotaxonomy (Diefendorf et al. 2019), support the interpretation that the biomarker composition of early gymnosperms has remained broadly consistent with that of extant taxa at the family level since at least ~250 million years ago (Leslie et al. 2012), particularly when assessed within sedimentary matrices (Diefendorf et al. 2019).

Moreover, evidence suggests that lipids and lignin co-evolved in lycophytes in response to environmental pressures such as irradiance, desiccation, and herbivory (Laskar et al. 2012; Pittermann 2010). Given that chemotaxonomic markers associated with extant lycophytes extend back to at least 300 million years ago (DiMichele et al. 1992, 2001), we interpret that

the gross molecular composition of lycophyte-derived (i.e., herbaceous) organic matter across the Paleozoic–Mesozoic transition is comparable to that of their modern counterparts.

To address the Reviewer's concern and clarify this interpretation, we have revised the manuscript accordingly (Lines 205-215). The data points in Figure 3D represent a mixing line between herbaceous and gymnospermous sources, which accounts for their clustering and is consistent with the transitional vegetation described in the text (Lines 192-194) and is back by modelling results (Figure 4).

Diefendorf et al. 2019. A phylogenetic analysis of conifer diterpenoids and their carbon isotopes for chemotaxonomic applications, *Organic Geochemistry*, 127, 50-58.

DiMichele, W.A., Aronson, R.B., 1992. The Pennsylvanian-Permian vegetational transition: a terrestrial analogue to the onshore-offshore hypothesis. *Evolution* 46 (3), 807–824.

DiMichele, W.A., Pfefferkorn, H.W., Gastaldo, R.A., 2001. Response of Late Carboniferous and Early Permian plant communities to climate change. *Annu. Rev. Earth Planet. Sci.* 29 (1), 461–487.

Laskar, D., Ke, D.D., Zeng, J., Gao, J., Chen, X., 2012. Py-GC/MS as a Powerful and Rapid Tool for Determining Lignin Compositional and Structural Changes in Biological Processes. pp. 9.

A.B. Leslie, J.M. Beaulieu, H.S. Rai, P.R. Crane, M.J. Donoghue, S. Mathews Hemisphere-scale differences in conifer evolutionary dynamics *Proceedings of the National Academy of Sciences*, 109 (2012), pp. 16217-16221

Pittermann, J., 2010. The evolution of water transport in plants: an integrated approach. *Geobiology* 8 (2), 112–139.

H.S. Rai, P.A. Reeves, R. Peakall, R.G. Olmstead, S.W. Graham Inference of higher-order conifer relationships from a multi-locus plastid data set *Botany*, 86 (2008), pp. 658-669

8. Palynology studies inferring vegetation change are mentioned (line 198) but not cited and no data is presented. As such it is difficult to comment on if there is a correlation or not between the datasets and what this means or does not mean for wildfire dynamics. Are the authors inferring a statistical or stratigraphic correlation?

We thank Reviewer 2 for highlighting the lack of references at the end of this paragraph. In response, we have added multiple citations relevant to the Early Triassic record in the Svalbard region (Lines: 212-214). Additionally, we would like to note that literature addressing global vegetation change is already cited at the beginning of the paragraph (Lines: 192-134). To further support the reader and strengthen the context, we have added an additional panel see Figure 2I, incorporating data compiled from these newly cited sources to visually illustrate the documented changes in vegetation.

9. The model work doesn't feel integrated into the paper I think this is because the wider aspects of the study and wildfire dynamics are not fully discussed in the introduction. Line 214-215 suggest that the figure presented as Figure 3B is evidence supportive of decreasing temperature impacting on precipitation and vegetation. This

seems at odds with the the data presented in the figure and the description within the legend.

We appreciate the reviewer's comment and agree that the manuscript would benefit from a clearer explanation of the modeled climate states and their implications for biome distribution and fire dynamics. In response, we have revised the introduction to provide more context on the climate modeling approach and to better frame the broader environmental changes discussed throughout the paper (see revised Lines 107-110).

We also added further explanation of the shift in biomes and climate states in the results and discussion sections. we have expanded the discussion to address the distinction between short- and long-term fire activity. As noted, models used to predict short-term fire dynamics (e.g., Pimont et al., 2021; Prichard et al., 2023) focus on ignition probability and available fuel, whereas long-term fire activity must also account for biomass turnover through net biome production. In this context, we argue that a transition to a cooler, drier climate state could still result in greater total biomass burned over time, particularly given the modeled increase in seasonal precipitation intensity near the study site. (See Lines 221–228 and 238–245).

To help guide the reader, we have also considered how best to distribute this information between the introduction, discussion and method (Lines 330-336). Some framing related to fast climate change and climate state transitions has been introduced early in the manuscript to better explain the complexity of fire–climate–biome interactions in a long-term context.

10. Given that the authors present data from one section and it isn't clear how these data map onto the broader area at this time interval or if the fire activity they have detected is substantive or not the discussion of broader influences of wildfires (Line 225-237) should be softened.

We thank Reviewer 2 for this observation. We agree that, given our data are derived from a single section, caution is warranted in extending interpretations to broader regional or global scales. In response, we have softened the language to better reflect the limitations of our dataset and have removed the final sentence, which may have overstated the broader implications of wildfire activity during this interval. Lines 253-268

11. The final paragraph of the paper talks about a topic (PyC) not previously mentioned and this seems like a rather strange way to close out a paper.

Although we do not explicitly discuss pyrogenic carbon (PyC) through most of the paper, in the absence of human influences on petrogenic carbon, PAHs would be a major component of PyC (Bird et al. 2015). Pyrogenic carbon cycle is now more explicitly addressed in the Introduction, helping to better integrate the manuscript and highlight the broader overarching questions it seeks to address.

Past research furthermore suggests that wildfire might have contributed up to ~20% of 'fresh' pyrogenic carbon across the Paleozoic-Mesozoic transition (Scott and Glasspool 2006) and furthermore could have remobilized large amounts of preexisting pyrogenic residue (Santin et al. 2013), which carries carbon cycle consequences commensurate with those of today (Bird et al. 2015, Hanke et al. 2017). This in mind, we decided to extend the somewhat disciplinary focus of this paper to the wider geosciences by linking (paleo)wildfire dynamics to recent and future carbon cycling.

Bird, M. I., Wynn, J. G., Saiz, G., Wurster, C. M. & McBeath, A. (2015). The Pyrogenic Carbon Cycle. *Annual Review of Earth and Planetary Sciences* 43, 273-298.

Hanke, U. M., et al. (2017). "What on Earth Have We Been Burning? Deciphering Sedimentary Records of Pyrogenic Carbon." *Environmental Science & Technology* 51(21): 12972-12980.

Santín C., Doerr SH., Preston C., Bryant R. (2013). Consumption of residual pyrogenic carbon by wildfire. *Int. J. Wildland Fire* 22::1072–77

Scott AC., Glasspool IJ. (2006). The diversification of Paleozoic fire systems and fluctuations in atmospheric oxygen concentration. *PNAS* 103::10861–65

Reviewer #3 (Remarks to the Author):

Overview:

The authors present PAHs and alkanes records spanning Smithian-Spathian substages, ca. 250 million years ago. The source of PAHs and the geographical extent of paleo-wildfire were discussed in detail. The manuscript present a novel interpretation of wildfire intensities shifted together with changing vegetation and precipitation patterns, as well as regional organic carbon cycle.

However, the discussion about relations between wildfires and climatic conditions need to be more focused to improve readability and logicity for readers. The method should be more detailed to enhance the reader's understanding of the dataset and results. Both figures in text and supplemental figures require minor revision. Please see specific comments below.

We have taken these suggestions on board and have made revisions accordingly. In the first instance, we have expanded the detail for our methods discussion with especial focus on information about biomarker separations (through flash chromatography) to make them more straightforward for our readers to understand.

Specific comments:

1. Line 75: Please add some description about the origin of PAHs and their climatic significance during the mass extinction events if word count allows.

This information is addressed in these lines 75-77. If there is something the reviewer specifically wants to have added to the text, please let us know what exactly. We are open to further input if word count allows.

2. Line 82: It is better to refer the stages or conodont zones here, so that the reader can compare between different sections.

There are latitudinal differences between ammonoid and conodont zones. To make it more globally understandable to a broad audience the authors have added the substage terminology in brackets. (Line 85)

3. Line 113: Please give specific values for the CAI, Ro% and MPI, and then discuss the thresholds for maturity (i.e. which value for immature/mature) to improve the reading interest of non-professional readers.

We thank the Reviewer for their comment. In response, we have added specific values for CAI, TAS, and Tmax to provide a clearer picture of the thermal maturity of the section. We have also included a discussion of the relevant thresholds for these parameters and expanded the background literature to support the readership in understanding how these values inform our interpretations. These additions are intended to strengthen the methodological transparency and contextual grounding of our study. (Lines: 118-121)

4. Line 115: In many publications, the plot Pr/nC17 vs. Ph/nC18 suggested the origin of organic matter or OM type, and the authors could cite published papers instead of manuscript under review. Pr/nC17 and Ph/nC18 ratios are also thought to decrease with increasing thermal maturity. However, this relationship only holds if the decomposition of the side chain of chlorophyll-a is the only or at least the predominant source of these isoprenoids. Please discuss in detail that why the plot Pr/nC17 vs. Ph/nC18 can be the indicator of maturity and what type of OM in this study.

We agree with the Reviewer that the interpretation of Pr/nC17 vs. Ph/nC18 ratios carries certain caveats. These limitations and considerations are thoroughly discussed in Blattmann et al. (2025, *Geochimica et Cosmochimica Acta*), which we explicitly reference in the manuscript. Given the length constraints and the primary focus of our study on Early Triassic wildfire activity, we have chosen to keep this section concise. We believe this approach provides readers with a clear reference for further detail while maintaining the focus and clarity of the manuscript.

5. Line 121: There are points in Figure 2E representing values of 0 (or nearly?) that should not be include in the graph. Please provide more reasonable interpretation of PAHs profile.

Thank you for raising this important point regarding data presentation. We struggled with how to present values that fall beneath our instruments' limit of detection (LOD) because such values are not necessarily zero. The lower member of our stratigraphic log contains numerous 'very low' concentrations (viz. less than 10% of the maximum downcore value) that might look like a zero value in our figures. We chose to retain these low values to show the full range of measured PAHs, which we believe is important for understanding downcore trends and potential variations in wildfire activity. The exact concentrations are available in the accompanying data repository. See SwissUBase Ref. 20755 <https://doi.org/10.48657/cdjz-sj28>

6. Line 123: Please include an interpretation of why the Pristane/n-C17 and phytane/n-C18 ratios can indicate petrogenic input. Additionally, please add this proxy to Figure 2 to illustrate the changes in petrogenic input across the section.?

Please see authors response in Reviewer 3 comment 4.

7. Line 129: Petrogenic input can be assessed by changes in alkyl PAHs, with various indicators available to quantify alkyl PAHs for evaluating petrogenic input.

To better assess potential petrogenic input, we have included a plot of the Methylphenanthrene Index (MPI-1) versus depth (Figure 2H). This provides an additional line of evidence for evaluating petrogenic versus pyrogenic PAH sources in the record.

8. Line 134: delete e.g. and cite publications directly.

This has been adjusted. Line 142

9. Line 139: Please consider another possible source that PAHs produced by regional wildfire and transport via aerosols.

To the best of our knowledge, we have considered and discussed the primary natural sources of PAHs relevant to a pre-Anthropogenic setting. These include petrogenic input, thermal degradation/ oil migration, volcanism, soil erosion and wildfire. All are addressed in detail in Lines 117–137 and 139-167 of the manuscript. If the reviewer has specific additional sources in mind, we would be open to further suggestions.

10. Line 142: “PAH levels” specifically refers to which indicators in Figure 2, and it seems that the figure does not match the description.

Thank you for pointing this out. We have now clarified the sentence to specifically reference each dataset clearly to ensure consistency between the text (Line 151) and the figure.

11. Line 152: The source of wildfire is discussed by excluding potential sources; please provide more concrete and direct evidence.

There is an important philosophical question nested within this comment that bears some discussion! Although we agree with the reviewer that direct or more robust evidence about Triassic wildfires would be very powerful. However, as noted in the manuscript (Lines 84-88), such evidence (i.e., Charcoal gap) does not exist and is absent in this time interval.

Consequently, we rely on deductive reasoning to exclude potential sources of PAHs as we can confidently exclude a source based on available data. Thus, while some of the data we use in our reconstructions might be open to reinterpretation, data-based exclusion of sources nonetheless holds because logical deduction still functions as a robust hypothesis-testing process in science.

Kleinhans, M.G., Buskes, C.J. and de Regt, H.W., 2010. Philosophy of earth science. *Philosophies of the sciences: a guide*, pp.213-236.

Lawson, A.E., 2000. The generality of hypothetico-deductive reasoning: Making scientific thinking explicit. *The American Biology Teacher*, 62(7), pp.482-495.

Johnson, M.E., 2010. Tracking Silurian eustasy: Alignment of empirical evidence or pursuit of deductive reasoning? *Palaeogeography, Palaeoclimatology, Palaeoecology*, 296(3-4), pp.276-284.

12. Line 162: The first sentence of this paragraph mentions biogeochemical cycles, please add explanations about impacts on biogeochemical cycles to increase the manuscript's readability by creating connections between the first sentence and subsequent paragraphs.

We acknowledge that the mention of "biogeochemical cycles" in the first sentence of the paragraph was not clearly integrated with the rest of the paragraph, which may have caused confusion for the reader. To improve clarity and coherence, we have revised the paragraph by removing the phrase from this early reference. Instead, we now address the impacts on biogeochemical cycles in greater detail later in the manuscript, where they are more directly supported by the context.

13. Line 164: The interpretation of LMW and HMW could add in supplementary table 2.

Diagnostic values have been added to Table 2 for all listed ratios.

14. Line 171: LMW is very common in Phanerozoic strata, so it is also considered as petrogenic input.

In the section "Thermal diagenetic overprint and petrogenic input" (Lines 117-137) petrogenic PAH origin is discussed in detail and higher amounts of petrogenic carbon is excluded in the Vendomdalen Member (Spathian).

We would like to clarify that this paragraph focuses on transportation pathways and wildfire extent using previously proxies established in previous studies (e.g. Karp et al 2020). The first sentence has been reworded in order to avoid confusion to the readership (Lines 179-180).

15. Line 172: What does "distinct trends between LMW and HMW" mean?

We thank the Reviewer for pointing out the unclear wording. This has now been revised for clarity, and a corresponding figure reference has been added to better support the readership and guide interpretation of the relevant data. (Line: 180-182)

16. Lines 173-180: The explanation of high LMW PAHs and low HMW PAHs is somewhat vague. Generally, the formation temperature or geothermal temperature may not be sufficient to produce HMW PAHs. However, the loss of HMW PAHs during long-distance transport could be one possible explanation. It is better to interpret PAHs transportation in conjunction with conclusive evidence or publications about increased wildfire activity in a large region, along with the wind systems that transport these PAHs to the study area.

As discussed in the manuscript (see charcoal gap discussion in introduction), there is currently a lack of data on wildfire activity for this time interval. This is what makes our record noteworthy, as it captures changing wildfire activity during the Early Triassic in conjunction with climate shifts. Given the age of the sediments (~250 million years) there is a lack of

correlated terrestrial outcrops, which could link specific fire events to PAH transport remains limited.

However, based on MITgcm modeling (Ragon et al 2024), the dominant winds at the latitude of our study site were likely westerlies. To help clarify the source-to-sink dynamics, we now indicate the relevant hinterland area in Figure 4 (square with sides of 1500 km), which reflects the approximate region from which material could have been transported to the site. A clarifying sentence has also been added to the caption of Figure 4 (Line 613).

Ragon, C., V  rard, C., Kasparian, J. et al. Alternative climatic steady states near the Permian–Triassic Boundary. *Sci Rep* 14, 26136 (2024). <https://doi.org/10.1038/s41598-024-76432-8>

17. Line 180: Please add wildfire evidence or sites in the larger region and how they transport to the study area.

The authors would like to clarify that this study reconstructs wildfire activity from approximately 250 million years ago using a marine sedimentary section, which captures signals from terrestrial wildfire occurring in the hinterlands. As discussed in the manuscript (Lines 80-114), the Early Triassic is an epoch characterized by the so-called "charcoal gap" as macro-charcoal is absent.

This study is the first of its kind to use PAHs as a proxy for wildfire activity during this interval, offering a new geochemical approach where traditional proxies are lacking. While we recognize the value of linking to terrestrial wildfire evidence in the broader region, such terrestrial sections are lacking for this region and difficult to correlate with marine records at the necessary temporal resolution.

For these reasons, we focus our interpretation on the PAH record preserved in the marine sediments, which we believe offers the most reliable insight into wildfire activity in this context.

18. Line 184: Vegetation types exhibit significant regional characteristics. Please emphasize in this paragraph that changes in regional vegetation types can indicate drier climate. Excepting changes in vegetation type, are there other evidences (i.e. thorns or mud crack) that suggest drought conditions?

This study is based on a marine sedimentary section, and no nearby terrestrial sections that can be confidently and precisely correlated stratigraphically are available for this region. Terrestrial sections of this age are generally scarce and, where they do exist, are difficult to correlate with marine records at the temporal resolution required for this type of paleoenvironmental reconstruction.

Regarding the suggestion about thorns: these are features associated with angiosperms, which did not evolve until the Cretaceous. The time interval studied here is the Early Triassic, well before the appearance of angiosperms, and thus such traits are not relevant to the vegetation or fire ecology of this period.

To support the discussion of vegetation change and its environmental implications, we have added additional information and references documenting floral turnover at the SSB and the associated aridification of the climate in the Spathian (Line 193-194).

19. Line 189: Many studies have examined the relationships among fire, climate, and vegetation; however, it would be valuable if the authors could offer a new perspective on these three aspects.

We appreciate this comment, as it highlights the need to further emphasize key concepts in our manuscript. While numerous studies have investigated the relationships among fire, vegetation, and climate in the Cenozoic; the origin and evolution of wildfire dynamics, particularly in the Paleozoic and Mesozoic remain poorly understood (e.g., Hamad et al. 2012; Baker 2022). This uncertainty is especially relevant during the rise of woody gymnosperms (Cressler 2001), which produced vanillyl-rich lignin (Busch et al. 2019). Compared to syringyl-containing biomass, vanillyl lignin may significantly influence fire behavior due to differences in organic matter degradation kinetics and burn intensity (Certini et al. 2011; Merino et al. 2015; Chen et al. 2022). Further complicating this picture are the non-analogue climate conditions of the Triassic (Mays 2022), which obscure our understanding of biomass-fire interactions during this critical interval, despite their relevance for modeling future climate-fire feedbacks (Heilmeyer 2019). See Lines 197 -203 in the manuscript.

Baker, S.J., 2022. Fossil evidence that increased wildfire activity occurs in tandem with periods of global warming in Earth's past. *Earth-Science Reviews*, 224, p.103871.

Certini, G., Nocentini, C., Knicker, H., Arfaioli, P. and Rumpel, C., 2011. Wildfire effects on soil organic matter quantity and quality in two fire-prone Mediterranean pine forests. *Geoderma*, 167, pp.148-155.

Chen, H., Wang, J.J., Ku, P.J., Tsui, M.T.K., Abney, R.B., Berhe, A.A., Zhang, Q., Burton, S.D., Dahlgren, R.A. and Chow, A.T., 2022. Burn intensity drives the alteration of phenolic lignin to (Poly) aromatic hydrocarbons as revealed by pyrolysis gas chromatography–mass spectrometry (Py-GC/MS). *Environmental science & technology*, 56(17), pp.12678-12687.

CRESSLER III, W.L., 2001. Evidence of earliest known wildfires. *Palaios*, 16(2), pp.171-174.

Hamad, A.M.A., Jasper, A. and Uhl, D., 2012. The record of Triassic charcoal and other evidence for palaeo-wildfires: signal for atmospheric oxygen levels, taphonomic biases or lack of fuel?. *International Journal of Coal Geology*, 96, pp.60-71.

Heilmeyer, H., 2019. Functional traits explaining plant responses to past and future climate changes. *Flora*, 254, pp.1-11.

Mays, C. and McLoughlin, S., 2022. End-Permian burnout: The role of Permian–Triassic wildfires in extinction, carbon cycling, and environmental change in eastern Gondwana. *Palaios*, 37(6), pp.292-317.

Merino, A., Chávez-Vergara, B., Salgado, J., Fonturbel, M.T., García-Oliva, F. and Vega, J.A., 2015. Variability in the composition of charred litter generated by wildfire in different ecosystems. *Catena*, 133, pp.52-63.

20. Line 193: In Figure 2, there was also an increase at about 70 m depth, and why this increase not be included in the discussion?

Thank you for pointing this out. This has been added to the text in order to provide a complete interpretation (Line 207).

21. Line 195: Please point out the timing of vegetation transition and the appearance of the retene in the profile.

As noted in the manuscript, retene is present throughout the section (Line 212). The timing of the vegetation transition is described in detail in Lines 192–196. To improve clarity and guide the reader, we now explicitly reference this transition again in Line 213, in direct connection with the DMP trend. This addition reinforces the link between biomarker signals and floristic changes during the Early Triassic.

22. Line 197: Are the molecular evidence and vegetation shifts consistent over time? At what specific time or depth in the profile do they align? Clarifying this will help the reader understand the sequence of events and the relationships among fire, climate, and vegetation.

We acknowledge the importance of understanding the alignment between molecular and palynological evidence of vegetational shifts. However, due to uncertainties related to taphonomic biases, it is challenging to determine precise temporal relationships between these proxies. Additionally, the time resolution of the sedimentary record is not well constrained, which limits our ability to confidently assign specific depths or horizons to synchronous events.

However, we now clarify in the manuscript that the simulated shift from a hot to a cold state is consistent with a change in major biomes, which supports our interpretation. This biome transition aligns with molecular evidence presented in the study and reinforces the patterns discussed in response to points 7–9 from Reviewer 2.

23. Line 210: Is there fossil evidence about the increase in biome diversity locally? If so, please cite them.

We acknowledge that the original wording was unclear and may have implied the existence of direct fossil evidence for increased biome diversity at the local scale. We apologize for this poor choice of words. The sentence in Line 225-227 has been revised for clarity.

24. Line 215: Figure 3B is refer to the source analysis of PAHs, please re-cite figure.

Thank you for pointing out the mistake. This has now been corrected.

25. Lines 225-236: This part is speculative and lacks precise, direct evidence regarding nutrient and carbon cycles. I suggest shortening this part and merging it with the model analysis above to enhance the completeness of the article.

We appreciate this thoughtful suggestion. We agree that the section previously lacked precision, and in response, we have shortened and refocused it to better reflect the current limits of the available evidence. We now clearly frame this part of the discussion in terms of

broader, open questions regarding nutrient and carbon cycling that remain to be addressed in future studies. However, we have chosen not to merge this section with the preceding discussion on biomes and climate, as the themes and focus of the two sections differ. We feel that maintaining them separately allows for greater clarity and thematic coherence.

26. Line 245: Wildfires are thought to be a major source of carbon dioxide released by forests into the atmosphere. Large and intensive wildfires can lead to forest ecosystems shift from carbon sinks to carbon sources. However, carbon sequestration mechanism in study area seems to lack of direct evidence. Additionally, the article does not address the frequency of fires, making it difficult to definitively conclude that carbon sequestration is occurring. It may be more appropriate to conclude that wildfires can have a significant impact on the global carbon cycle, although further data and models are needed to fully explain this impact.

We agree with the suggestion that, while wildfires can play a major role in the global carbon cycle, more data and modeling are required to fully assess their impact in this context. In response, we have reworded and toned down the final sentences of the paragraph to avoid overinterpretation and better reflect the limitations of the available evidence. We now emphasize that the study does not provide direct evidence of carbon sequestration via wildfire activity in the region. Instead frame our conclusions more cautiously in line with the reviewer's recommendation. (Line 277-286)

27. Line 252: The method should be more detailed to enhance the reader's understanding of the dataset and the identity of results.

This is a valuable suggestion, and we have acted upon it accordingly! More specifically, we have expanded the detail for our methods discussion with information about biomarker separations (through flash chromatography) to make them more straightforward for our readers to understand (Lines 282-288).

28. Line 254: Please add the paleogeographic latitude and longitude, regional climate conditions and section description in the method. If publications have detailed descriptions of the section, please cite them in the method.

More information on the section and regional geology has been added to the methods parts (Lines 282-288) along with supporting citations.

29. Line 260: Please describe the maturity proxies and the results of RockEval in detail. It is recommended to put the maturity proxies and results in the method part or supplementary file.

As requested in Reviewer 3 comments 3 and 4 more information on the thermal maturity proxies have been added to the "Thermal Diagenetic Overprint and Petrogenic Input" section of the discussion along with the supporting references from previous studies. The results of the RockEval are added as table 3 to the supplementary materials (See Line 617 as well as file Table 3). References have been added to the RockEval pyrolysis to further support the readership on thermal maturity.

30. Line 280: Please add description of the compound identification and references for the PAHs identification in supplementary file.

We have updated our molecular analysis section of the Materials and Methods (c.f., Lines 280-328) to include more comprehensive information including peer-reviewed literature references about the approaches we used for PAH identification, quantification and uncertainty propagation in the main text. This information is now also mentioned in our supplementary file.

31. Line 284: Please add one paragraph about detailed steps of the model and a description of the model results.

Given space constraints, we have opted to include only a brief summary of the modeling setup in the main text (Lines 330-336). We now clarify that the BIOME4 model was coupled offline to simulated climate fields from the MITgcm in order to determine the most likely biome at each terrestrial location. This information has been added to the introduction to provide clearer context for readers. For those interested in the full modeling procedure, we now refer to the cited references in the manuscript.

Figures :

32. Figure 2: Please add the significance of the arrows on the horizontal axis. add the lithology legend.

The lithology is now described in figure caption with reference to the Blattmann et al 2024 paper that is focused on stratigraphy (Line 566-568). The arrows are described in the figure caption, and we have abstained from adding our interpretation to the figure so that the reader can make a more unbiased conclusion.

33. Figure 3: Please add a significance P-value test.

This has now been added. Please see reviewer 2 comment 6.

34. Figure 4: Please label A, B, C that match with text.

Thank you for pointing this out. This has now been corrected.

35. Supplementary Figure 1: Please enlarge the horizontal and vertical coordinates.

This figure has been removed from the paper as it is now available in Blattmann et al. (2025) GCA. Please see Reviewer 3 Comment 4.

36. Supplementary Figures 2-4: Please add identification of PAHs. The peak of $m/z = 276$ could not be identified in the figure so it is necessary to describe the identification in the text.

See Reviewer 3 comment 30. This information has been added to the Figure captions See lines 595-612. Also more information is now written in the Methods section.

37. Supplementary Figure 5: The 0 value could not be shown in the profile, especially HMW PAHs profiles.

See response to Reviewer 3 Comment 5.

All line numbers refer to lines in the tracked changes manuscript.

REVIEWERS' COMMENTS:

Reviewer #2 (Remarks to the Author):

Thank you for responding to my comments and providing the detailed response letter outlining how you have responded to questions raised in my initial review of your paper. I think some of the text within your response letter needs to be within the body of the paper.

For example the discussion on how you handled the data to detect pulses of PAHs is missing from the text. This needs to be included.

We have added a dedicated subsection in the Methods (Lines 348-354) outlining how data were handled. Specifically, we now describe the use of change-point analysis to detect thresholds in our datasets, identifying pulses as statistically significant shifts distinct from background variability. We also note the limitation posed by the lack of comparable Early Triassic PAH records for longer-term context.

If you are basing the analysis on shifts from the section average I think it might be worth thinking about replotting your data as difference from the average this would highlight the nature of the PAH spike.

We appreciate this suggestion, which prompted extensive discussion among the authors of this manuscript. While we considered applying statistical transformations, we ultimately decided against this approach, as it might overemphasize PAH excursions and unintentionally bias interpretation. Instead, we believe that presenting the raw values allows readers to directly evaluate the changes in PAHs and their implications for wildfire activity without overinterpretation. As a compromise, we now also present averaged PAH values in Figure 2D and 2E, which helps highlight significant differences while preserving transparency in the underlying dataset (see revised Figure 2).

Another minor example of information that would enhance the readability of the document is the reason why the the studies on line 89 are controversial.

The sentence has now been adjusted to contain information explaining the reason for the controversy (see Lines 95-99).

The introduction still feels a little bit underdeveloped and that is still impacting on readability. The modelling still feels more like a bolt onto the ms rather than an integral part of the study. I would strongly encourage the authors to take advantage of the greater word limit for Communications Earth & Environment to provide a more extensive set up the paper and provide a more detailed discussion of some of the relationships you have found.

We have added further information and supporting references on the modeling approach in both the Introduction (Lines 116-121) and the section Biome and Climate Change (various points Lines 210-260). We hope these additions provide clearer context and better support for the reader.

For example I think it would be beneficial to explore in more detail the mixing model presented on lines 205-208. The authors highlight in the response this is backed up figure four but again this isn't articulated in the paper.

Additional information on dimethylphenanthrene ratios with supporting references has been incorporated into the manuscript to support the reader (see Lines 207–213).

I think many of the issues raised by reviewer three and responded to in the cover letter would also be better folded into the revised ms rather than being in the response.

We agree with the reviewer that several of the points raised are best addressed directly in the manuscript rather than solely in the cover letter. Accordingly, we have incorporated additional information on DMP plots, the ambiguity surrounding the charcoal gap, data analysis and trend identification, as well as integration with modeling results into the revised manuscript.

I recognise that the structuring of a paper is rather subjective but I'm not sure having the main results of the paper being presented in the introduction (figure 2) is the logical way to structure the paper. The main purpose of the citing of the figure in the introduction is to cite the sedimentary log. If the authors wish to do this wouldn't it be better to combine the log with the location map to have one complete introduction figure and then have the actual data embedded in the results section?

To improve the flow of the manuscript and maintain a clearer separation between the Introduction and Results, we have removed the reference to Figure 2 from the Introduction and relocated the relevant sentence to the Materials and Methods section. This change keeps the figures simple and straightforward to understand and avoids redundancy while more strictly separating introduction and results (See lines 110-112 and 292-301).

Reviewer #3 (Remarks to the Author):

The manuscript has been revised according to the comments and has been reorganized to improve its logic and completeness. Based on the author's response, there are still four comments that require further clarification in the text:

1. The response to comment 2 (Line 85) still needs to specify the biostratigraphy. If there are regional differences, it should be indicated which regional biostratigraphy belongs to and the corresponding international stratigraphic age. Additionally, if word count allows in the methods section, some information on stratigraphy and sedimentology could be added, as ref. 24 and 62 do not present absolute chronostratigraphy or stratigraphy research.

We do agree with Reviewer 3 that more information on the biostratigraphy would be helpful to the readership. Line 85, however, refers to a compilation of literature (references 15 and 16). We have now specified the biostratigraphic framework used for the Stensiöfjellet section (see Methods and Materials lines 298-301). This includes the regional scheme and how it corresponds to other international stratigraphic ages. We also note differences in defining the SSB. In addition, we have added supporting information and references on the regional geology and stratigraphy to provide more context for the reader (see Methods and Materials lines 293-297).

2. The response to comment 5 (Line 121): Can I understand that these very low values are actually background values? If so, the author could consider using the detection limit as the background value. This would make the discussion of PAH concentrations as evidence of fire more reasonable.

This is a really interesting comment because it highlights the 'art' of figures and their interpretation. Although we agree with the reviewer that identifying a background/baseline signal for each PAH would be useful for highlighting the occurrence of fire, it also implies we are confident that the instrumental detection limit is commensurate with the aforementioned baseline signal. As well, this approach implies we think that low values represent the natural baseline. Given the relatively coarse sampling resolution and the paucity of other regional PAH records, we are hesitant to make this assertion. For instance, what if the natural baseline signal for some PAHs is an intermediate concentration, whereas others are essentially absent (depending on, e.g., delivery mechanism)?

3. The response to comment 9 (Lines 117–137 and 139-167): The biological source is also possible source, especially LWM PAHs. Considering the need for rigor, this possibility needs to be ruled out, e.g.

Krauss, M., Wilcke, W., Martius, C., Bandeira, A. G., Garcia, M. V., & Amelung, W. (2005). Atmospheric versus biological sources of polycyclic aromatic hydrocarbons (PAHs) in a tropical rain forest environment. *Environmental Pollution*, 135(1), 143-154.

Daisy, B. H., Strobel, G. A., Castillo, U., Ezra, D., Sears, J., Weaver, D. K., & Runyon, J. B. (2002). Naphthalene, an insect repellent, is produced by *Muscodora vitigenus*, a novel endophytic fungus. *Microbiology*, 148(11), 3737-3741.

Although studies on its biological origin are modern researches, it cannot be completely ruled out that ancient organisms are not a source of PAHs in the strata.

We thank the reviewer for pointing this out and bringing up the discussion. We would like to clarify that low-molecular-weight PAHs such as naphthalene, emphasized in both suggested papers (Krauss et al. 2005 and Daisy et al 2002), are not presented in this manuscript. Naphthalene cannot be detected, which likely is most likely due to both the age of the samples and the low melting point of this molecule. Given the absence of naphthalene data in our record and its lack of discussion in the manuscript, we do not consider it appropriate to include the discussion of its biogenicity. Similarly, while Krauss et al. (2005) report phenanthrene in angiosperm vegetation from tropical regions, this is not directly relevant here, as our study interval predates the rise of angiosperms and the site is located outside the tropics.

We acknowledge that the origin of perylene remains debated, with some studies suggesting links to fungal activity or termite mounds (Grice et al., 2009; Hanke et al., 2019; Krauss et al., 2005). For this reason, we chose from the outset not to emphasize perylene in the manuscript text. We also note that there is currently insufficient studies on fungal activity during this epoch (e.g., limited fungal spore data), which prevents a robust comparison to test such interpretations. To maintain the manuscript's focus on wildfire activity, we have therefore excluded detailed discussion of perylene. However, the full dataset is provided in the supplementary material (Figures S1–S3) and in the data repository, ensuring accessibility for future studies that may wish to further investigate the biogenicity of this particular compound.

Hanke, U. M., Lima-Braun, A. L., Eglinton, T. I., Donnelly, J. P., Galy, V., Poussart, P., ... & Reddy, C. M. (2019). Significance of perylene for source allocation of terrigenous organic matter in aquatic sediments. *Environmental Science & Technology*, 53(14), 8244-8251. DOI: 10.1021/acs.est.9b02344

Kliti Grice, Hong Lu, Pia Atahan, Muhammad Asif, Christian Hallmann, Paul Greenwood, Ercin Maslen, Svenja Tulipani, Kenneth Williford, John Dodson, New insights into the origin of perylene in geological samples, *Geochimica et Cosmochimica Acta*, Volume 73, Issue 21, 2009, <https://doi.org/10.1016/j.gca.2009.07.029>.

Krauss, M., Wilcke, W., Martius, C., Bandeira, A. G., Garcia, M. V., & Amelung, W. (2005). Atmospheric versus biological sources of polycyclic aromatic hydrocarbons (PAHs) in a tropical rain forest environment. *Environmental Pollution*, 135(1), 143-154.

4. The response to comment 11: I agree that the excluding potential sources in this ms is logical deduction. However, if the process of elimination is needed to determine the source of PAHs, all possible sources of PAHs should be excluded, as mentioned above.

We appreciate the reviewer's concern and would like to clarify the use of diagnosis by exclusion in this study. This approach is commonly applied in fields such as medicine, where the true number of possible causes is not always known in advance. The aim is not to claim that the remaining explanation is definitively correct, but rather that it is the explanation most consistent with the available evidence once others are systematically ruled out. In our case, we use this method to exclude unlikely PAH sources, leaving wildfire as the explanation that best fits the observed data, while remaining open to future findings that could refine or challenge this interpretation.